# Multilayer Diamond Coatings Applied to Micro-End-Milling of Cemented Carbide

**DOI:** 10.3390/ma14123333

**Published:** 2021-06-16

**Authors:** Eduardo L. Silva, Sérgio Pratas, Miguel A. Neto, Cristina M. Fernandes, Daniel Figueiredo, Rui F. Silva

**Affiliations:** 1Department of Materials and Ceramics Engineering, CICECO—Aveiro Institute of Materials, University of Aveiro, 3810-193 Aveiro, Portugal; elsilva@ua.pt (E.L.S.); sergiopratas@ua.pt (S.P.); mangelo@ua.pt (M.A.N.); 2Palbit S.A., P.O. Box 4, Branca, 3854-908 Albergaria-a-Velha, Portugal; cfernandes@palbit.pt (C.M.F.); dfigueiredo@palbit.pt (D.F.)

**Keywords:** micro-end-milling, CVD diamond, multilayer coating, cemented carbide, transmission electron microscopy

## Abstract

Cobalt-cemented carbide micro-end mills were coated with diamond grown by chemical vapor deposition (CVD), with the purpose of micro-machining cemented carbides. The diamond coatings were designed with a multilayer architecture, alternating between sub-microcrystalline and nanocrystalline diamond layers. The structure of the coatings was studied by transmission electron microscopy. High adhesion to the chemically pre-treated WC-7Co tool substrates was observed by Rockwell C indentation, with the diamond coatings withstanding a critical load of 1250 N. The coated tools were tested for micro-end-milling of WC-15Co under air-cooling conditions, being able to cut more than 6500 m over a period of 120 min, after which a flank wear of 47.8 μm was attained. The machining performance and wear behavior of the micro-cutters was studied by scanning electron microscopy and energy-dispersive X-ray spectroscopy. Crystallographic analysis through cross-sectional selected area electron diffraction patterns, along with characterization in dark-field and HRTEM modes, provided a possible correlation between interfacial stress relaxation and wear properties of the coatings. Overall, this work demonstrates that high adhesion of diamond coatings can be achieved by proper combination of chemical attack and coating architecture. By preventing catastrophic delamination, multilayer CVD diamond coatings are central towards the enhancement of the wear properties and mechanical robustness of carbide tools used for micro-machining of ultra-hard materials.

## 1. Introduction

Cobalt-cemented tungsten carbides (WC-Co) are known for their combination of high hardness, strength, and toughness, along with excellent dimensional stability at high temperatures. These characteristics make WC-Co ideal for the fabrication of precision parts such as molds and cutting tools [1]. Nevertheless, such mechanical properties also make WC-Co difficult to machine, in particular when high dimensional accuracy is required. Although machinability can be improved from the side of the workpiece [2], the cutting process plays a fundamental role in the optimization of cutting efficiency and surface integrity. Processes such as EDM (electric discharge machining), grinding, and polishing are commonly used but exhibit a low material removal rate and can lead to the formation of surface microcracks [3]. Micro-milling is a microfabrication process that enables the production of high aspect ratio precision components. As the diameter and cutting edge radius of the tool are reduced, the dynamics of machining are significantly changed, and material removal from the workpiece through chip formation can cease and give way to ploughing (elastic/plastic deformation without material removal), particularly if machining parameters such as the feed per tooth are not correctly adjusted [4,5,6]. As tool wear and cutting edge radius increase, burr formation is commonly observed, and cutting forces can increase severely along with fatigue effects, often times leading to catastrophic failure of the tool shaft [4,5]. The performance of cemented carbide micro-end mills, coated with CrTiAlN for micro-milling of Ti6Al4V, was investigated by Xia and co-workers [7]. Under conventional micro-milling conditions, the coated microtool exhibited tool nose breakage, coating spalling, and adhesive wear, after 1500 mm length, leading to reduced surface quality and tool life.

CVD (chemical vapor deposition) diamond coatings are widely recognized by their extreme hardness and wear resistance, thus being employed for improving tool life in micromachining operations. Lei and co-workers [8] demonstrated a 10-fold durability increase of carbide micro-ball end mills when coated with microcrystalline diamond (MCD), for graphite machining. Wu and Cheng [9] investigated the applicability of nanocrystalline diamond (NCD)-coated micro-end mills to the machining of Al6061-T6. The authors concluded that NCD can lead to a 16% reduction in cutting forces and a 25% reduction in tool wear, as well as to a decrease in burr formation. Torres and collaborators [10] observed a similar tendency during micro-end-milling of Al6061-T6 as well. WC-Co micro-end mills were coated with sub-microcrystalline diamond (SMCD) and NCD for comparison, using hot filament-assisted CVD. Besides lower wear and improved durability, a reduction of 90% in cutting forces was measured for SMCD-coated tools, as compared to the uncoated ones. Although a lower cutting force reduction was registered for the NCD-coated tools, this type of coating led to less delamination failure [10]. Yan et al. [11] compared between single-layer NCD and MCD coatings, as well as multilayer MCD/NCD, for milling of natural marble. Their findings included improved adhesion of the MCD coatings and higher resistance to crack propagation in NCD coatings. The alternate combination of both morphologies in the same coating led to an increase in service life as well as improved surface quality [11].

Over many years of research and industrial development, it became clear that the machining performance of diamond-coated WC-Co cutting tools can be optimized by adjusting the lifecycle of the coating in four different levels: (i) WC-Co grade selection and surface conditioning, (ii) CVD growth parameters, (iii) coating design, and (iv) selection of machining parameters. Wei and co-workers investigated the influence of different pre-treatments on the microstructure and adhesion of CVD diamond grown on top of WC-Co with varying cobalt content of 3%, 6%, 10%, and 13%. Comparison was made between pretreating WC-Co substrates via chemical attack with Murakami’s and Caro’s (MC) reagents and via boronization with borax (Na_2_B_4_O_7_·10H_2_O) powder at 950–1000 °C. The authors concluded that chemical attack of WC-Co was efficient for low cobalt content (3%), while for higher cobalt content, the surface exhibited deep heterogeneous pits [12]. In contrast, Pratas et al. [13] demonstrated that the MC chemical attack on WC-Co must be optimized for each carbide grade. By studying different attack times and Murakami’s dilutions, the authors showed that WC-7Co substrates can be prepared by the MC attack, leading to pit-free surface roughening and homogeneous diamond coatings deposited by HFCVD. Furthermore, the produced diamond coatings were able to withstand loads up to 1250 N on Rockwell C indentation testing without significant delamination [13]. Almeida et al. [14] used a variant of the MC chemical treatment, with Aqua Regia instead of Caro’s reagent for the removal of cobalt in submicrometric carbide substrates. The authors demonstrated that the diamond seeding and nucleation processes are more effective, allowing a greater adhesion of the film to the carbide tool, with interfacial crack resistance of 1.6 kgf/µm^2^ [14]. Using the same pre-treatment, the authors compared uncoated carbide with NCD (nanocrystalline diamond)-coated carbide bits for drilling of pre-sintered carbide (WC-5.5Co) [15]. The uncoated bit registered an increase in axial force from 4 to 20 N after 4 holes, with a low feed rate of 20 mm/min. The NCD-coated carbide tool allowed feed rates of 940 mm/min, reaching the same axial force of 20 N for a feed rate of 500 mm/min [14]. 

Wang and collaborators [16] investigated the cutting performance of diamond coatings with different morphologies, applied to high-speed milling of graphite. It was shown that coatings with MCD/NCD multilayer design exhibited the best cutting performance, as well as machining efficiency, and surface finishing of the graphite workpieces [16].

This work expands on a previously developed study [13], where the importance of adapting chemical pre-treatments specifically to each carbide grade was highlighted. Such optimized pre-treatment was used in the present investigation to prepare diamond-coated carbide micro-end mills applied to machining of fully sintered carbide parts. The multilayer architecture of the diamond films was characterized by TEM (transmission electron microscopy) and correlated with the performance and wear mechanisms of the coated tools. To the authors’ knowledge, this is the first time the fine structure of CVD diamond multilayer coatings has been analyzed by TEM from the standpoint of cutting performance as applied to micro-milling of ultra-hard materials. This report demonstrates the importance of WC-Co preparation methods towards the maximization of micro-milling cutters via coating with CVD diamond multilayer coatings. 

## 2. Materials and Methods

WC-7 wt.% Co micro-ball end mills (Palbit, S.A., Albergaria-a-Velha, Portugal) with a shaft diameter of 2.5 mm were preconditioned for cobalt etching and surface roughening, as reported elsewhere [13]. In short, WC grains were attacked with diluted Murakami reagent (10 g KOH + 10 g K_3_Fe(CN)_6_ + 150 mL water) during 22.5 min for surface roughening. Subsequently, the tools were immersed in H_2_SO_4_:H_2_O_2_ 1:14 during 3 s, for cobalt etching. The pre-treated samples were seeded with diamond powder and coated with multilayer diamond films in a hot filament-assisted CVD (HFCVD) reactor. Tungsten filaments (⌀ = 150 μm) were used for gas activation and the reaction chamber exhibited a cylindrical geometry, with water cooling. Before the growth stage, the filaments were carburized during 30 min at 2300 °C and a CH_4_/H_2_ ratio of 0.05. After the carburization step, growth conditions were employed, and the films were composed of nanocrystalline diamond (NCD) and sub-microcrystalline diamond (SMCD) according to the parameters in Table 1. One of the end mills was used for TEM characterization, while the others were used for machining of sintered carbide. The tool for TEM was coated with three layers, as indicated in Table 1, for 14 h. The tools for machining were coated with an identical layered structure, for a longer period of 45 h in total. 

The adhesion of the diamond coatings was evaluated by Rockwell C indentation tests performed by a Zwick/Roell Z020 machine (ZWICKROELL, Ulm, Germany) with a Brale-type diamond indenter. Three indentations per load were recorded, in order to determine the critical load.

The diamond-coated micro-end mills were tested for machining a hard metal workpiece composed by WC-15 wt.% Co with 3 µm WC grain size, using a CNC miller (CNC Makino IQ300, MAKINO, Tokyo, Japan), with air cooling (Figure 1). The following cutting conditions were used: Cutting speed (V_c_) = 54.8 m/min, spindle speed (n) = 30,000 rpm, axial depth of cut (a_p_) = 0.03 mm, radial depth of cut (a_e_) = 0.03 mm, feed rate (f_z_) = 6 µm/tooth.

The microstructural characterization of the coated tools was performed by scanning electron microscopy (SEM, Hitachi SU-70, HITACHI, Tokyo, Japan) and energy-dispersive X-ray spectroscopy (EDS, Bruker Quantax 400, BRUKER, Billerica, Massachusetts, United States). A lamella of the micro-end mill with thinner coating was extracted via focused ion beam (FIB, FEI Helios 600 Dual Beam, liquid gallium ion source, 30 kV, FEI, Hillsboro, Oregon, United States). The thin lamella was then characterized by transmission electron microscopy (TEM, JEOL 2010F FEG-TEM, 200 kV, JEOL, Tokyo, Japan), in high-resolution mode (HRTEM), selected area electron diffraction (SAED), and dark-field mode. DigitalMicrograph 3 software (Gatan Microscopy Suite^®^, GATAN, Inc., Pleasanton, CA, USA) was used for processing of HRTEM imagery. 

## 3. Results and Discussion

WC-7Co micro-ball end mills (Figure 2a), with WC average grain size of 0.8 µm (Figure 2b) and 1730 HV10, were coated with alternating NCD/SMCD diamond layers, as shown in Figure 2c. The design of the coating encompasses a nucleation layer of NCD, followed by a transition layer with increasing crystallite size, culminating in the SMCD layer located in the middle section of the coating, approximately. Above the SMCD layer, the grain size gradient is reversed, with crystallite size diminishing progressively until the top NCD layer (Figure 2c), completing a total of 11.5 μm in average thickness.

The main purpose of diamond coatings when applied to WC-Co tools is to provide wear resistance and remove heat away from the cutting edge. To do so, the films must resist delamination failure, as well as chipping along the cutting edge. Carbide grades have a strong influence on the behavior of diamond coatings since they share the interface and provide support. Chae et al. [17] investigated the effect of the grain size of diamond films for coating of heat-treated WC-Co cutting tool inserts. Heat treating of WC-Co promoted the increase of WC grain size on the surface, leading to an increase in roughness as well. The main purpose of this treatment was to contribute to higher diamond film adhesion. By comparing microcrystalline (SMCD) and nanocrystalline (NCD) diamond coatings, the authors concluded that the larger grain size of SMCD led to the formation of interfacial porosity due to inability of the diamond crystals to conform to the surface topography of WC-Co. By means of N_2_ addition to the gas phase, diamond grain size was reduced, and NCD was grown instead of SMCD. With this morphology, surface coverage was increased and porosity was eliminated, leading to higher adhesion and interfacial toughness [17]. In the present study, nanocrystalline diamond was the morphology of choice, during the initial growth stage, aiming at lower interfacial stress and higher surface coverage. Figure 3 shows the TEM characterization of a diamond film structurally identical to the one shown in Figure 2c, but with lower average thickness of 3.4 µm. While the layered structure of the film is visible in bright-field TEM, dark-field imaging using a (111) spot clearly shows a smooth transition gradient between the nucleation and the SMCD layers. Conversely, the transition between the SMCD and the top NCD layer is more pronounced. The crystallographic analysis of the diamond layers in Figure 4 shows multiple SAED projections acquired across the thickness of the film. These provide a clearer understanding of the fine structure of the film. The nucleation layer is comprised of randomly oriented nanocrystals, similar to the top NCD layer. This is confirmed by the ring patterns from selected areas a1–a3 and b2–b3, which can be assigned mainly to (111), (220), and (311) diamond planes.

On the SMCD layer (Figure 4b), Bragg reflection spots (selected area b1) can be observed instead of rings, confirming the existence of larger diamond crystals.

The adhesion of the coatings was evaluated by Rockwell C indentation with a Brale-type conical diamond indenter. The results are shown in Figure 5, where absence of delamination was observed for indentation loads up to 1250 N, demonstrating the high adhesive strength of the coating. As highlighted in [13], this result reinforces the importance of optimizing cobalt etching for each specific carbide grade. Accordingly, the diamond coating was able to withstand indentation loads comparable to those of other reports, where much more elaborate interfacial engineering was employed, namely by the use of interlayers and boronization [12,18,19]. 

The diamond-coated micro-end mills were tested for machining of WC-15 wt.% Co at a cutting speed of 54.8 m/min and a feed rate of 6 µm/tooth, under air-cooling conditions. Operating at a spindle speed of 30,000 rpm, the micro-end mill was able to cut 6576 m, lasting 120 min before reaching a flank wear VB of 47.8 µm (Figure 6a). Although a standardized tool life criteria similar to ISO 8688-2:1989 [20] does not exist for micro-end-milling, the tool was still in a viable cutting condition. Abrasion and adhesion were identified as the predominant wear mechanisms (Figure 6b). This is further depicted in Figure 7a, where abrasive grooves are visible along the flank face. In the same area, it was possible to identify the presence of attached WC-Co from the workpiece on top of the coating, as analyzed by EDS (Figure 7b,c). This is in accordance with the findings in several reports, where abrasive and adhesive wear were commonly observed [21,22]. Further analysis of the diamond coating reveals that despite evidence of coating detachment in both the flank and rake faces, extensive delamination did not occur. This shows the effectiveness of the diamond coating in protecting the main body of the tool, and was associated with the multilayer design, which enabled a progressive layer-wise wear (Figure 6b) of diamond instead of catastrophic delamination failure of the coating and loss of wear protectiveness. This design was also linked to the absence of chipping along the cutting edge, which is commonly observed for carbide tools during intermittent cutting operations such as end-milling [23]. Accordingly, as shown in Figure 6b, material loss from the coating was more severe on the top NCD layer, which is visibly retracted in comparison to the remaining layers of the coating underneath. 

For comparison, Suwa and co-workers have conducted cutting experiments with carbide ball end mills coated with CVD diamond of similar thickness (10 µm) [24]. When machining an identical carbide grade (WC-15Co, 3 µm WC size), the tool lasted only 68.1 m, using coating peel-off as tool life criteria [24]. Wang et al. [16] presented a thorough study focusing on the tribological properties and comparison of the cutting performance of diamond coatings with different morphologies: MCD, SMCD (sub-microcrystalline diamond), NCD, and MCD/NCD multilayer. All coatings presented identical thickness of ~9 µm and were applied on WC-6Co end mills and used for high-speed milling of graphite in the absence of cutting fluids. Comparison with uncoated carbide end mills was also presented. It was found that MCD exhibits much higher adhesion strength than SMCD and NCD coatings. MCD/NCD multilayer coatings exhibited the lowest friction coefficient and highest wear resistance, yielding the best cutting performance. Flank wear of ~100 µm was reached after 120, 560, 720, 800, and 960 m of cutting length for the uncoated WC-Co-, NCD-, SMCD-, MCD-, and MCD/NCD-coated tools, respectively [16]. 

The improvement of machining efficiency through the use of a MCD/NCD multilayer architecture has been demonstrated by several authors [25,26]. Salgueiredo and collaborators [27] have presented an analytical model showing that in multilayer diamond coatings, delamination depends on the stress at the coating/substrate interface. The maximum value of the von Mises parameter J_2_^1/2^ is reached at the MCD/NCD layer transitions, and cracks tend to propagate along these regions, providing a toughening mechanism and delaying the coating delamination [27]. 

In the present report, the interface between the carbide tool and the diamond coating was analyzed by TEM, in order to establish a possible correlation with machining efficiency. High-resolution TEM of the interfacial region revealed the presence of diamond crystallites with (111) plane orientation (Figure 8). Figure 9 depicts this interfacial region in dark-field mode, obtained from the diffraction spots of diamond (111) and (220) planes. It is clear that a large fraction of diamond nanocrystals is located at the interface. Furthermore, there is a very high surface coverage and conformation of WC by diamond nanocrystals, as shown by the absence of voids.

Such interfacial structure was associated with the high adhesion of the coatings and high durability of the micro-mills. When diamond coatings are grown on top of substrates with high thermal dilation mismatch, as is the case of WC-Co, thermally induced stresses can determine premature failure of the coating and tool. The high density of grain boundaries in the interfacial NCD layer, along with the apparent high diamond/non-diamond carbon ratio, most likely enabled relaxation of the film. This contributes to a better accommodation of thermally induced stress at the interface between the diamond coating and the tool, as well as enhanced crack deflection at the interface between diamond layers, thus preventing catastrophic delamination and extending tool durability. 

Hence, this report demonstrates the effectiveness of multilayer diamond architecture in ensuring high durability of carbide tools in micro-end-milling of ultra-hard materials such as sintered carbide workpieces. Furthermore, structural correlation between the performance and the microstructure of the coating shows that high adhesion and wear resistance can be linked to a combination between high fraction of diamond content for stronger bonding with the substrate, and high density of grain boundaries for improved stress relaxation at the interface with WC-Co.

## 4. Conclusions

WC-7Co micro-end mills were coated with multilayer diamond by HFCVD and used for machining of sintered WC-15Co. The substrates were chemically pre-treated for cobalt etching and surface roughening. The coatings exhibited high adhesion, withstanding a Rockwell C indentation load up to 1250 N.

After 120 min of machining, the tools were still in usable condition. Such tool life largely surpasses the performance of diamond coatings available in the literature for similar microtool geometries. Abrasion and adhesion were identified as the predominant wear modes. The effectiveness of the multilayer architecture in preventing catastrophic coating delamination was demonstrated by the layer-wise progressive wear of the diamond coating.Characterization by TEM was presented for the first time for CVD diamond multilayer coatings applied to micro-milling carbide tools. The structure of the coating revealed that the interface between the carbide tool and the diamond coating is majorly composed of diamond nanocrystals. This high fraction of diamond in detriment of non-diamond carbon, along with high density of grain boundaries, most likely promotes stress relaxation, and increases the interfacial toughness of the coating. This could be correlated with the high adhesion of the coatings and good performance of the coated tools.

As follow-up for a previous publication [13], this work confirms the dependence of machining performance on the adjustment of surface conditioning to specific WC-Co grades, as well as on the design of diamond coating. Furthermore, the combination of NCD and SMCD layers was demonstrated as suitable for enhancing the wear properties and mechanical robustness of carbide tools used for micro-machining of ultra-hard materials.

## Figures and Tables

**Figure 1 materials-14-03333-f001:**
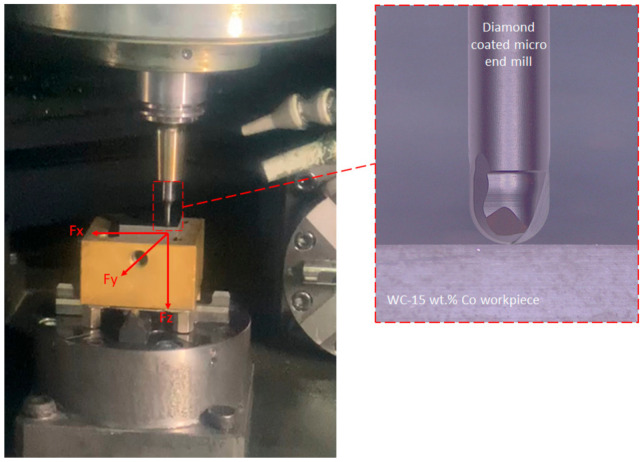
Setup used for machining of WC-15 wt.% Co with diamond-coated micro-end mill (inset).

**Figure 2 materials-14-03333-f002:**
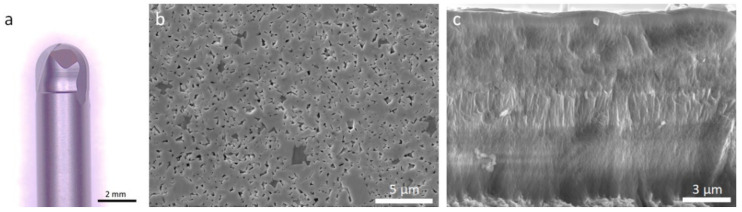
Diamond-coated end mill (**a**) and microstructural characterization of (**b**) the WC-7Co tool and (**c**) the multilayer diamond coating.

**Figure 3 materials-14-03333-f003:**
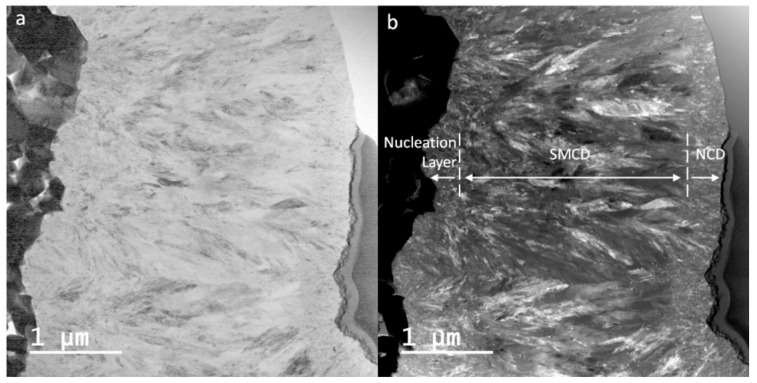
Transmission electron microscopy characterization of a multilayer diamond coating identical to the one used for the end mills in (**a**) bright-field and (**b**) dark-field modes.

**Figure 4 materials-14-03333-f004:**
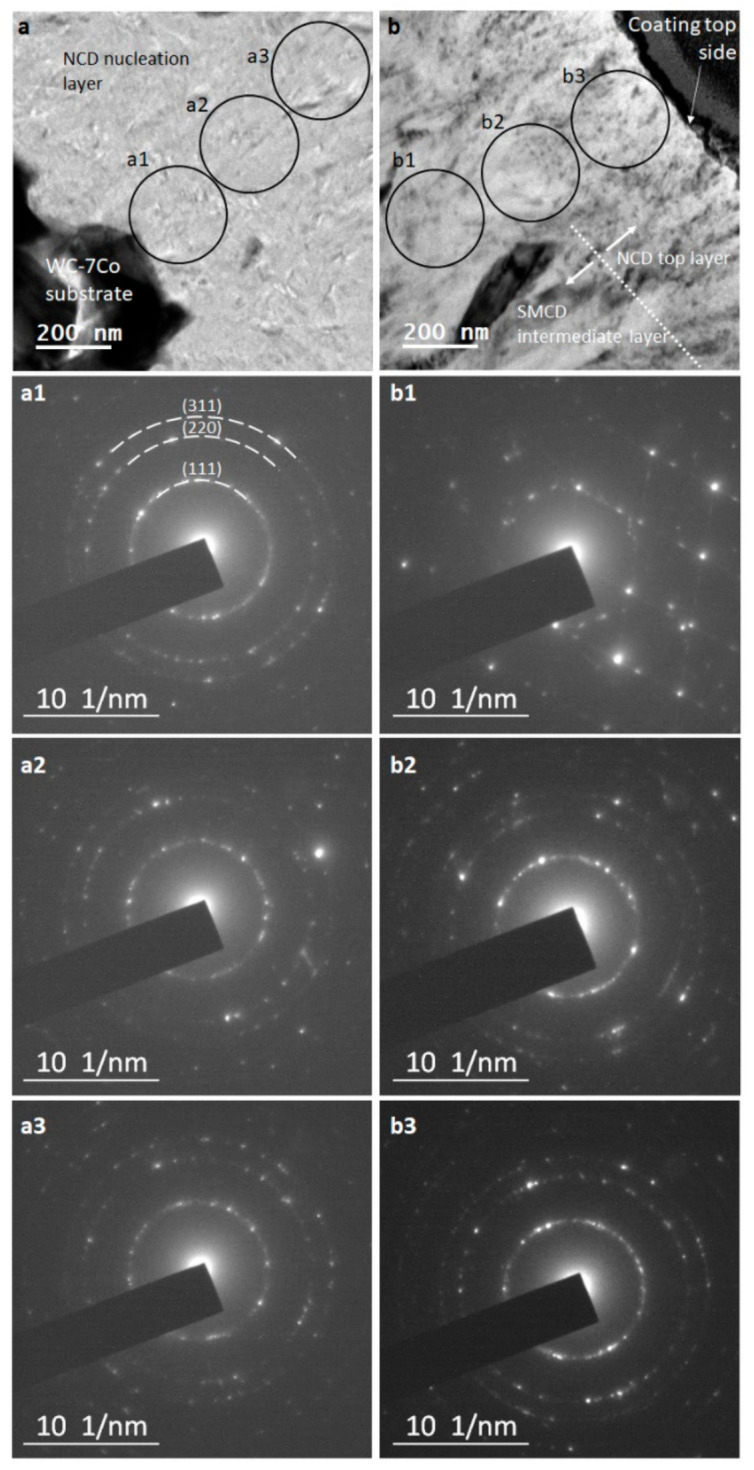
Selected area diffraction patterns of the multilayer diamond coating recording across its thickness in the following order from the substrate side (**a**) to the surface side (**b**): a1, a2, a3, b1, b2, b3.

**Figure 5 materials-14-03333-f005:**
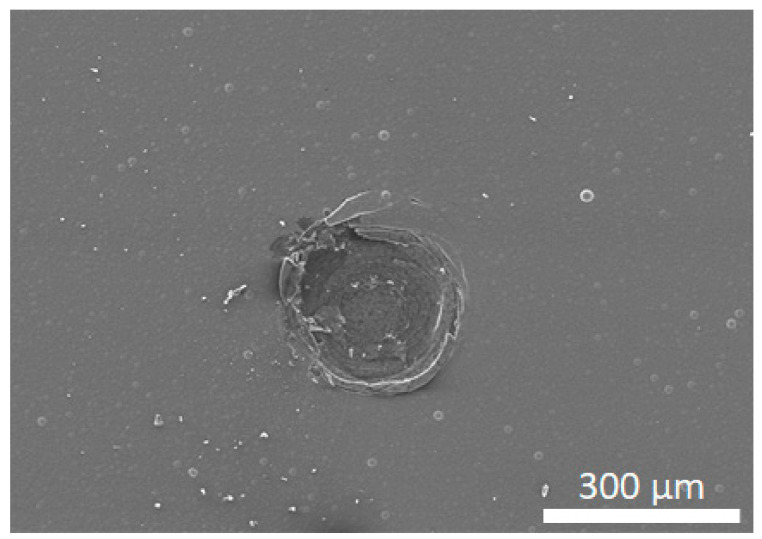
Adhesion testing of the coating by Rockwell C, with a load of 1250 N.

**Figure 6 materials-14-03333-f006:**
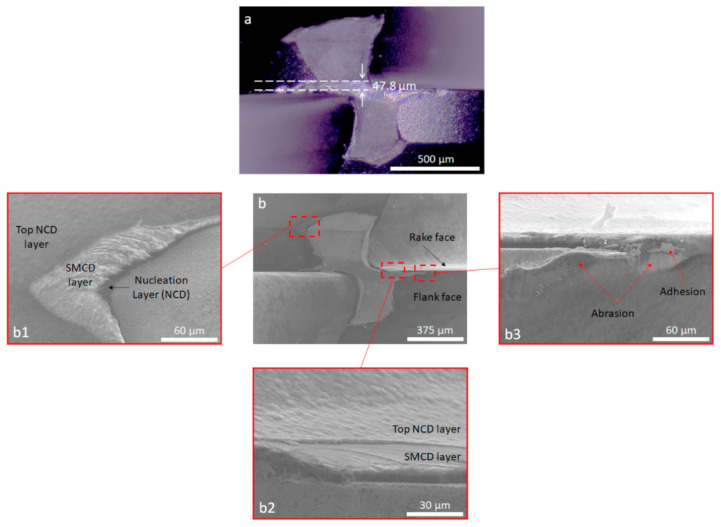
Worn surface of the diamond-coated ball end mill after machining of WC-15Co for 120 min. (**a**) Top view showing flank wear. (**b**) Top view in SEM showing progressive layer-by-layer wear enabled by the multilayer design of the diamond coating (b1, b2) and main wear mechanisms (b3).

**Figure 7 materials-14-03333-f007:**
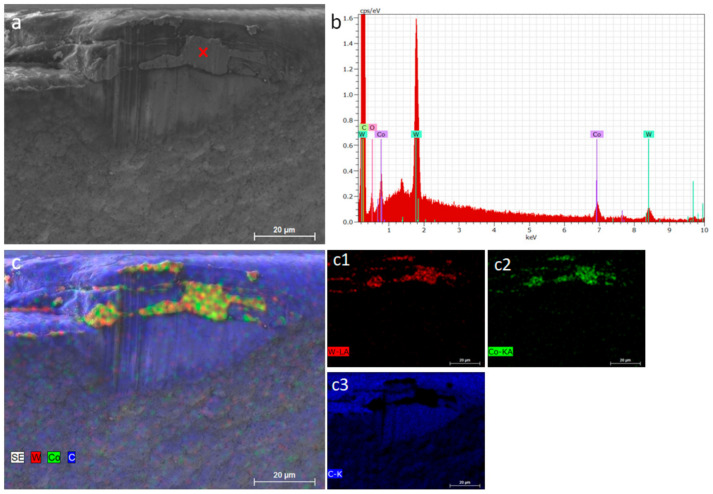
Wear characterization of the coating by energy dispersive X-ray spectroscopy (EDS) showing the spectrum (**b**) recorded from the red mark in (**a**). (**c**) The elemental mapping of image (**a**), along with individual maps (c1–c3), demonstrating adhesion of cemented carbide from the workpiece on the worn tool.

**Figure 8 materials-14-03333-f008:**
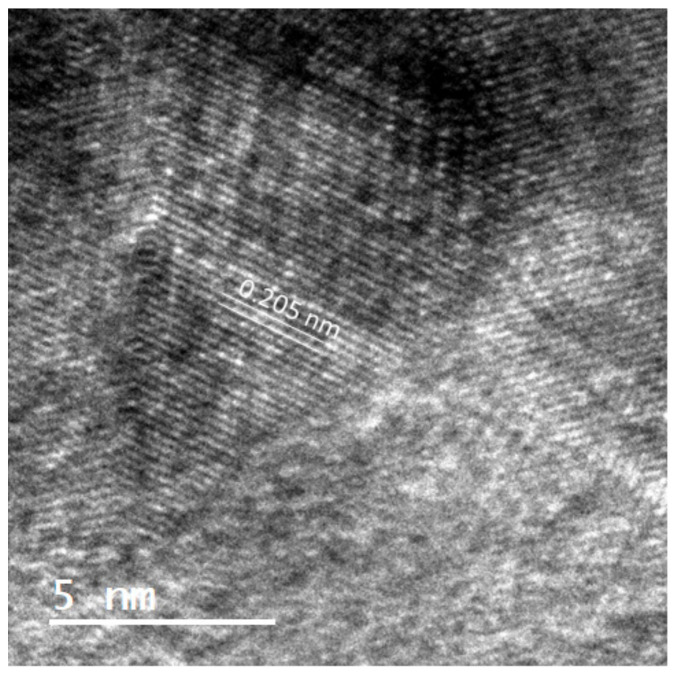
High-resolution TEM showing diamond crystallites with interplanar spacing of 0.205 nm, corresponding to the (111) plane orientation of diamond.

**Figure 9 materials-14-03333-f009:**
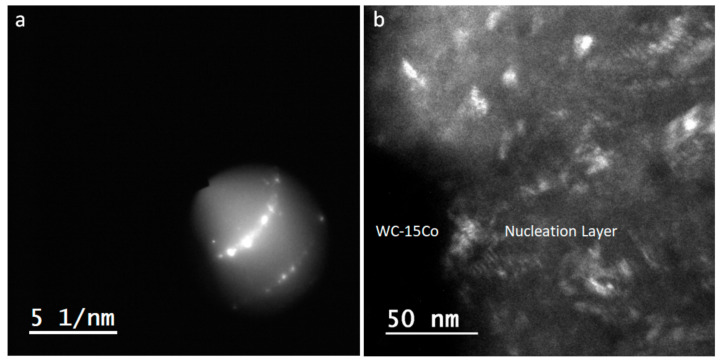
Dark field of the WC/diamond interface (nucleation layer) (**b**) recorded using the diffraction spots shown in (**a**).

**Table 1 materials-14-03333-t001:** Parameters used for diamond growth by HFCVD.

Layer	CH_4_/H_2_ Gas Flow Ratio	T_filament_ (°C)	T_substrate_ (°C)	Pressure (mbar)
1—Nucleation	0.03	2200	700	10
2—SMCD	0.02	2300	850	15
3—NCD	0.04	2250	800	10

## Data Availability

The data presented in this study are available upon request from the corresponding author.

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
