# Peer review of "Multilayer Diamond Coatings Applied to Micro-End-Milling of Cemented Carbide"

_materials, 2021, doi:10.3390/ma14123333_

Round 1
Reviewer 1 Report
The article entitled " Multilayer CVD Diamond Coatings Applied to Micro 2Machining of Cemented Carbide" is well written. The results are explained with proper supporting analysis. The theme of the article is that depositing nanodiamond film for increasing the strength of the machines is very important and crucial for modern technology. It is worth writing an article about it. Since, the article belongs to coating by CVD technique, so, author can provide more information on the CVD process.
Author Response
The authors appreciate the kind comments of the reviewer. The Materials and Methods Section was improved and now includes further details regarding the CVD process, as requested.
Reviewer 2 Report
The paper is generally written in clear and concise manner without obvious typographical errors.
The article has only one unclear segments. Figure 3 and the results of the SAED are not fully described in the article. Therefore, I recommend a better description and discussion of the results here.
One note: To better describe the damage after wear tests would fit the display surface before and after the test using AFM.
Author Response
Figure 3 and the part of Results and Discussion concerning Figure 3 have been improved in order to provide a better description and understanding of the results.
The authors are thankful for the suggestion of displaying surface before and after the test using AFM. Although such measurements are unpractical at this point, these will be considered in a future publication.
Reviewer 3 Report
The reviewer comments of the paper «Multilayer CVD Diamond Coatings Applied to Micro Machining of Cemented Carbide»- Reviewer
The authors presented an article «Multilayer CVD Diamond Coatings Applied to Micro Machining of Cemented Carbide». However, there are several points in the article that require further explanation.
Comment 1:
Title needs to be rewritten more specifically and clearly. Now from the title it is not clear why it is the article and it investigates? Title should reflect the purpose of the article.
Comment 2:
Abstract.
What are the quantitative and qualitative results obtained? What a scientific novelty and practical significance?
Abbreviations in the title are unacceptable. Moreover, if they first appear in the text of the article, they must be explained.
Comment 3:
The introduction of the article should be significantly improved.
Authors need to add a paragraph analyzing the relevance of Cemented Carbide. What is this material? What micromachining problems are there?
What are the “white” spots? Why is your research important to science and the reader?
Add article in the introduction:
Journal of Materials Research and Technology 2021. DOI: 10.1016/j.jmrt.2021.05.021
You must prove that scientists did not solve the problem of these studies before.
At the end of the introduction formulate a clear and understandable purpose of the article at the end of the introduction.
Comment 4:
- Materials and Methods
For devices and machine used in research, indicate in parentheses (manufacturer, city, country).
Give an experimental setup where you can see the workpiece, cutting tool, measuring devices. Sign what is where. In general, show everything in detail and describe in the text. Provide in the table the geometric parameters of the cutter (Main cutting edge angle, rake and clearance angles, rounding radius, etc.).
What cutting conditions are being investigated?
How many repetitions are used in the measurement? What statistical methods are used to process experimental results? Please describe this in more detail.
Are all the figures in the article original? If not needed appropriate citations and publisher permissions.
The quality and resolution of all figures needs to be improved.
At what point in time are measurements taken? How does this compare to tool life and wear?
Comment 5:
- Results and Discussion
What tool life do figures 2, 5 correspond to?
I would like to see the corresponding dependences tool life – wear.
As an example, refer to the article:
International Journal of Advanced Manufacturing Technology 2020, 107(1), 3511–3525.
DOI: 10.1007/s00170-020-05236-7
I would like to see in the text of the article an analysis of the physics of cutting for each figure.
Comment 6:
It will be useful to add a section of Nomenclature in which to sign all the physical quantities and abbreviations encountered in the article. There are many physical quantities in the text and such a section will help to find the description of the necessary element.
For example,
d : Shaft diameter (mm)
CVD : Chemical Vapour Deposition
etc.
Comment 7:
The conclusions need to be improved.
What is the novelty of the article? What is the practical significance? What are the differences from previous works?
Conclusions should reflect the purpose of the article.
Use style:
*Conclusion 1.
*Conclusion 2.
Etc.
The article is interesting. However, the article needs to be improved. Now there are many dubious questions, each of which should be answered in detail by the authors. Authors should carefully study the comments and make improvements to the article step by step. All changes should be highlighted in color. After major changes can an article be considered for publication in the "Materials".
Author Response
Comment 1:
"Title needs to be rewritten more specifically and clearly. Now from the title it is not clear why it is the article and it investigates? Title should reflect the purpose of the article."
The title has been modified accordingly, and is now more specific referring to micro end milling instead of micro machining.
Comment 2:
"Abstract: What are the quantitative and qualitative results obtained? What a scientific novelty and practical significance?"
The abstract has been modified to provide quantitative results and a clearer view of the scientific novelty, namely, the critical indentation load, the flank wear, cutting length and cutting time.
"Abbreviations in the title are unacceptable. Moreover, if they first appear in the text of the article, they must be explained."
The abbreviation in the title has been removed accordingly.
Comment 3:
"The introduction of the article should be significantly improved. Authors need to add a paragraph analyzing the relevance of Cemented Carbide. What is this material?"
A paragraph regarding cemented carbide and its characteristics was added at the beginning of the introduction.
"What micromachining problems are there?"
The micromachining problems were already described in the original manuscript. These can now be found in lines 50 to 58 of the introduction in the revised manuscript.
"What are the “white” spots? Why is your research important to science and the reader?"
The manuscript presents currently existing micro milling problems in the Introduction section, which are indicative of possible “white” spots. The presented CVD diamond-based solution to improve some of these problems, is demonstration of the importance of the current research. Nevertheless, the modifications proposed by the reviewer, particularly in the abstract, end of introduction and conclusions, have contributed to a better understanding of the importance of the presented research.
"Add article in the introduction: Journal of Materials Research and Technology 2021. DOI: 10.1016/j.jmrt.2021.05.021"
The referred article was cited in the introduction, it is reference [2].
"You must prove that scientists did not solve the problem of these studies before."
In order to prove this, reference to a micromachining related article from 2019 has been added from lines 61 to 65 of the revised manuscript.
"At the end of the introduction formulate a clear and understandable purpose of the article at the end of the introduction."
The paragraph at the end of the introduction has been expanded to provide a clearer explanation of the purpose of the article.
Comment 4:
"Materials and Methods: For devices and machine used in research, indicate in parentheses (manufacturer, city, country). Give an experimental setup where you can see the workpiece, cutting tool, measuring devices. Sign what is where. In general, show everything in detail and describe in the text. Provide in the table the geometric parameters of the cutter (Main cutting edge angle, rake and clearance angles, rounding radius, etc.)."
A new Figure 1 with the experimental setup is now included in the revised manuscript. The requested geometric parameters of the tool cannot be provided, since the manufacturer and partner in this project, Palbit, S.A., does not wish to disclose this information.
"What cutting conditions are being investigated?"
The cutting conditions were added in the Materials and Methods Section.
"How many repetitions are used in the measurement? What statistical methods are used to process experimental results? Please describe this in more detail."
The information regarding the used number of repetitions was added to the Materials and Methods section.
"Are all the figures in the article original? If not needed appropriate citations and publisher permissions."
All the figures in the manuscript are original.
"The quality and resolution of all figures needs to be improved."
The quality of all figures has been improved. All figures currently have a resolution of 300 dpi.
"At what point in time are measurements taken? How does this compare to tool life and wear?"
The measurements were taken after 120 minutes of machining time. However the tool was still in good machining condition, i.e., the tool life was higher than 120 minutes, but for practical reasons the measurements were interrupted at this point and total tool life was not determined.
Comment 5:
"Results and Discussion: What tool life do figures 2, 5 correspond to?"
As already explained in the original manuscript and captions, Figure 2 is characterization of the coating before machining and Figure 5 corresponds to 120 minutes of machining.
"I would like to see the corresponding dependences tool life – wear. As an example, refer to the article: International Journal of Advanced Manufacturing Technology 2020, 107(1), 3511–3525. DOI: 10.1007/s00170-020-05236-7. I would like to see in the text of the article an analysis of the physics of cutting for each figure."
The present manuscript is essentially focused on CVD diamond coatings and corresponding demonstration as a viable solution for improving tool life in micro end milling of ultrahard materials. The authors agree with the reviewer regarding the tool life-wear dependence. However, as this was not our central focus, such measurements have not been acquired with enough quantity and significance to present in the current manuscript. The authors are thankful for the suggestion and do intend to present a tool life-wear analysis in a future publication.
Comment 6:
"It will be useful to add a section of Nomenclature in which to sign all the physical quantities and abbreviations encountered in the article. There are many physical quantities in the text and such a section will help to find the description of the necessary element. For example, d: Shaft diameter (mm); CVD: Chemical Vapour Deposition; etc."
A nomenclature section was added at the beginning of the manuscript.
Comment 7:
"The conclusions need to be improved. What is the novelty of the article? What is the practical significance? What are the differences from previous works? Conclusions should reflect the purpose of the article. Use style: *Conclusion 1. *Conclusion 2. Etc."
The Conclusions section has been improved in terms of structure and content, as suggested by the reviewer
Round 2
Reviewer 3 Report
The authors have improved the article according to the comments. The article can be accepted for publication.